# Perceptual decisions are biased by the cost to act

Nobuhiro Hagura[1,2]*, Patrick Haggard[1], Jörn Diedrichsen[1,3]

[1]Institute of Cognitive Neuroscience, University College London, London, United Kingdom; [2]Center for Information and Neural Networks (CiNet), National Institute of Communications and Technology, Suita City, Japan; [3]Brain and Mind Institute, Western University, London, Canada

**Abstract** Perceptual decisions are classically thought to depend mainly on the stimulus characteristics, probability and associated reward. However, in many cases, the motor response is considered to be a neutral output channel that only reflects the upstream decision. Contrary to this view, we show that perceptual decisions can be recursively influenced by the physical resistance applied to the response. When participants reported the direction of the visual motion by left or right manual reaching movement with different resistances, their reports were biased towards the direction associated with less effortful option. Repeated exposure to such resistance on hand during perceptual judgements also biased subsequent judgements using voice, indicating that effector-dependent motor costs not only biases the report at the stage of motor response, but also changed how the sensory inputs are transformed into decisions. This demonstrates that the cost to act can influence our decisions beyond the context of the specific action.

*For correspondence: n.hagura@nict.go.jp

**Competing interests:** The authors declare that no competing interests exist.

## Introduction

In laboratory experiments, participants are often asked to make decisions that are purely based on the features of the sensory input – a process that we refer to here as perceptual decision-making. However, in many of our daily situations, decisions are made in a behavioural context, in which the action that follow our decisions can differ dramatically in terms of required physical effort (or the *motor cost*). For example, in the orchard, one may aim to pick the reddest-looking apple from the tree. Some of the apples may be hanging high-up on the tree, which will require more effort to pick compared to other fruits hanging on the lower branch. In such situations, does the difference in the motor cost between the options influence the decision of which fruit to pick? If so, is such influence a result of serial integration between the perceptual decision (i.e. decision based on the visual feature) and the motor decision (i.e. decision for action selection to avoid the effortful action) at the output stage, or is the perceptual decision itself is affected by the cost on the downstream action? It has been shown that physical effort is used in motor planning (*Huang et al., 2012*; *Izawa et al., 2008*), and the physical effort to obtain a reward can influence behavioural decisions (*Prévost et al., 2010*; *Hosokawa et al., 2013*). Moreover, the uncertainty in perceptual decisions is transmitted to the motor system, influencing the parameters of action control (*de Lange et al., 2013*). However, it remains unclear whether the motor cost is simply integrated with the perceptual decision to optimise the expected utility (*Burk et al., 2014*; *Cos et al., 2014*), or whether the preceding experience of unequal motor costs can recursively influence the perceptual decision itself. Here, we show that manipulating the motor response cost for arm movements during a visual motion discrimination task changes not only the decision when responding with the arm, but also when reporting the perceptual decisions verbally.

**eLife digest** Imagine you are in an orchard, trying to decide which of the many apples to pick. On what do you base your decision? Most research into this type of decision-making has focused on how the brain uses visual information – about features such as colour, size and shape – to make a choice. But what about the effort required to obtain the apple? Does an apple at the top of the tree look more or less tempting than the low-hanging fruit?

To answer this kind of question, Hagura et al. asked volunteers to decide whether dots on a screen were moving to the left or to the right. The volunteers indicated their choice by moving one of two levers. If they thought the dots were moving to the right, they moved a lever in their right hand. If they thought the dots were moving to the left, they moved a lever in their left hand. What the volunteers did not know, however, is that one of the levers was slightly heavier and therefore harder to move than the other.

Hagura et al. found that the volunteers biased their decisions away from the direction that would require the most effort. If the right-hand lever was heavier, the volunteers decided that dots with ambiguous motion were moving to the left. Those for whom the left-hand lever was heavier felt that the same dots were moving to the right. The participants showed this bias despite failing to notice that the levers had different weights. Moreover, they continued to show the bias even when subsequently asked to simply say their answers rather than use the levers.

These results indicate that the effort required to act on a decision can influence the decision itself. The fact that participants were biased even when responding verbally, and despite being unaware that the levers differed in weight, suggests that they were not deliberately choosing the easier option. Instead, the cost to act changed how they perceived the stimuli themselves. The findings also suggest that it might be possible to help people make better decisions by designing environments in which less favourable options require more effort.

# Results

First, in Experiment 1, we examined if the decision of the visual motion direction can be biased when one of the two responses requires more effort. Ten right-handed participants observed a moving random-dot stimulus and made decisions about the direction of motion (leftward or rightward) (*Britten et al., 1992*). Participants held two robotic manipulanda, one in each hand. They indicated their decision by either moving their left hand (indicating leftward decision) or right hand (rightward decision, *Figure 1A*). In the baseline phase, the resistance for moving the manipulanda was the same for both hands (velocity-dependent resistance: 0.10 Ns/cm). In the subsequent induction phase, the resistance for the left hand increased by a small amount each time the participant moved the left hand (0.0008 Ns/cm; *Figure 1B*). Because the change was gradual, most of the participants did not report becoming aware of the increased motor cost when asked afterwards, even though their left hand was eventually exposed to 1.8 times greater resistance than the right (0.18 Ns/cm for left, 0.10 Ns/cm for right; see Materials and methods and *Figure 1—figure supplement 1*). This procedure was employed to minimise any cognitive strategy participants may use, such as explicitly avoiding the costly hand response regardless of the decision about the visual stimulus. In the test phase, participant then continued to perform the visual discrimination task under the accumulated asymmetry in manual resistance. We plotted the proportion of rightward judgement against different stimulus intensities, and determined the point of subjective equality (PSE, the point at which participants judge 50% of the trials to go rightward) for both the baseline and the test phase (*Figure 1D*). If the increased physical resistance for expressing leftward judgements was incorporated into the decision, the proportion of 'leftward' judgements should decrease in the test phase compared to the baseline, resulting in the shift of PSE towards the left (*Figure 1D*). As expected, the PSE shifted towards the left from baseline to test phase (−4.33%, paired t-test (2-tailed): $t_9$ = 2.43, p=0.038, $d$ = 0.76, 8/10 individuals showed the effect) (*Figure 1D–E*, *Figure 1—source data 1*). This indicates that the participants started to avoid making motion direction decisions in which the response is costly.

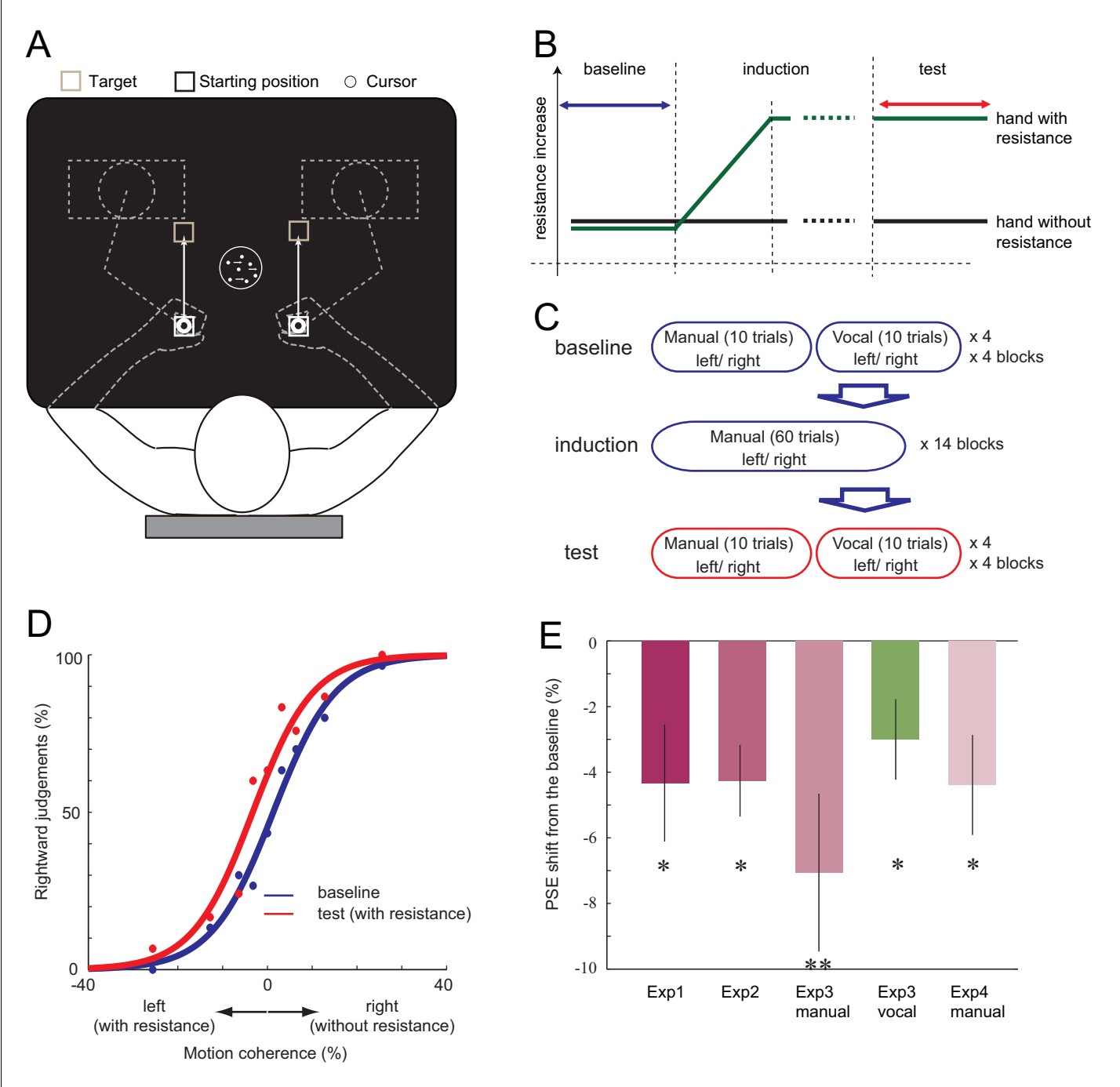

**Figure 1.** Setup of the experiment and the shift of PSE induced by the motor cost. (**A**) Participants made 15 cm reaching movement to the target with their hand (left or right), in response to the perceived direction (left or right) of the random-dot motion. (**B**) In all the experiments, the baseline phase and the test phase was interleaved by the induction phase, in which the resistance for one of the manipulandum movement gradually increased. (**C**) In Experiment 3, the baseline and the test phase included both manual and vocal motion discrimination, each being serially presented within a 10 trial block. (**D**) Fitted psychometric function to the probability of a response towards the right in the baseline (blue) and the test (red) phase of a representative participant (Experiment 1). Negative motion coherence value indicates the leftward motion (with manual resistance), and positive towards the right (without manual resistance). (**E**) Shift of PSE from the baseline in Experiments 1, 2, 3 and 4. Negative value indicates the PSE shift towards the motion direction with resistance (i.e. decreased judgements towards the motion direction having resistance in their manual response). Error bars indicate standard error of mean across participants. Data for *Figure 1E* is available as *Figure 1—source data 1*. *p<0.05, **p<0.01.

The following source data and figure supplement are available for figure 1:

*Figure 1 continued on next page*

*Figure 1 continued*

**Source data 1.** Individual PSE shift for Experiments 1–4, which is the data summarised in *Figure 1E*.
**Figure supplement 1.** Example of the peak resistive force increase profile in the study.

In Experiment 2, we examined whether motor cost and visual features need to be directly associated, or whether simply gaining experience of one action being more effortful than the other is sufficient to bias subsequent decisions. The baseline and the test phase involved judging direction of visual dot motion, as in Experiment 1. However, the induction phase was now replaced with a simple reaching movement, in which the participants moved their left or right hand according to a simple leftward or rightward arrow presented in the centre of the screen. As in Experiment 1, the resistance for moving the left hand was gradually increased. The motor cost during the induction phase was not associated with any motion direction judgement; the participants were only exposed to the gradually increasing motor cost differences between the two hands. Again, PSE significantly shifted leftwards in the test phase (*Figure 1E*, *Figure 1—source data 1*; baseline vs. test; mean $-4.26\%$, paired t-test (two-tailed); $t_8 = 3.91$, p=0.005, d = 1.3, 8/9 individuals showed the effect). This indicates that the direct association of higher motor cost with a specific decision during the induction phase is not critical for inducing the bias. This may suggest that the (implicit) knowledge about the response costs is sufficient to recursively influence the decision. Alternatively, these results could indicate that the bias is only transiently induced during the test phase itself.

In Experiments 1 and 2, we showed that manual motor costs reliably bias decisions that involve these manual response. Where in the process of translating a stimulus into a response does this bias arise? A simple model posits that decision-making occurs in three sequential stages (*Gold and Ding, 2013*). First, features of the sensory input are extracted and encoded as a sensory representation. Second, a categorical decision is made based on this sensory representation (decision layer). Third, the output from the decision layer is transmitted to the relevant effector for the response (*Figure 2*). One possibility is that the motor cost only biases the decision layer in the context of the specific response (*Oliveira et al., 2010*) (*Figure 2A*). In other words, the motor cost only influences decisions when the participant anticipates to perform the action associated with the motor cost, thus, the decision simply takes into account the upcoming motor costs. Alternatively, the repeated exposure to the manual motor cost may affect the perceptual decision about this type of stimulus in general, no matter which effector is used to make a response (*Figure 2B*) (*Bennur and Gold, 2011*; *Filimon et al., 2013*). Finally, the motor cost could also directly bias the sensory representation (*Figure 2C*), affecting the initial encoding of the information before it is transmitted to the decision layer. Only in the two latter scenarios, should the bias observed during the manual decisions generalise to decisions expressed with a different effector. In Experiment 3, we therefore examined whether a hand-specific motor cost could also influence a visual judgement that used a vocal response. A manual to vocal transfer of the motor cost effect would indicate that the motor cost influences the decision about the visual stimulus itself (i.e. perceptual decision), not the decision coupled with the effector selection (i.e. motor decision).

Fourteen new participants performed the visual motion discrimination as in Experiment 1. In the induction phase, we gradually increased the resistance for one of the hands while participants performed manual decisions as in Experiment 1. The resistance was increased for half of the participants (7) on the left hand, and for the other half on the right hand, accounting for any hand-dependent effects. To analyse these left and right resistance increase data together, we aligned the data depending on the side of the resistance applied by assigning negative motion coherence level to the motion direction associated with the direction of the resistance. During the manual task, participants moved their left or right hand according to their perceived motion direction. For the vocal task, participants indicated the direction of the motion by vocally responding 'left' or 'right' (*Figure 3—figure supplement 1A*) without moving their hands. During the baseline and the test phase, participants alternated between tasks: each 10 trials of manual judgements were followed by 10 trials of vocal judgements (*Figure 1C*). This 'top-up' procedure is commonly used to assess the effect of sensory adaptation on the subsequent perceptual judgements (*Fujisaki et al., 2004*). If the bias

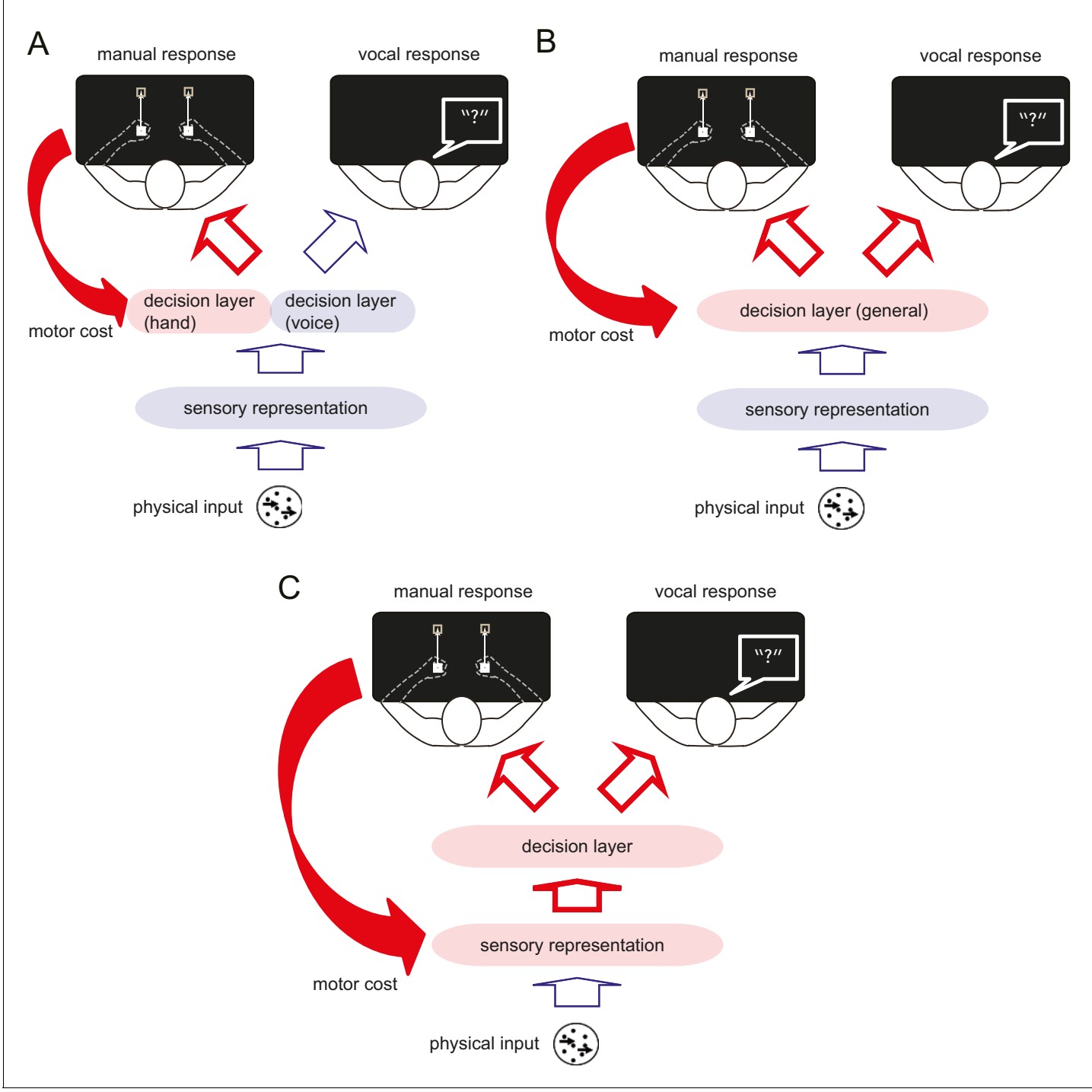

**Figure 2.** Schematic diagram illustrating the process of perceptual decision making, and the possible influence of the motor cost on the decision process. Perceptual decision making consists of three different processing stages. First, the features of the sensory input are extracted and encoded as in the sensory representation. Second, the perceptual (categorical) decision is made based on this sensory representation (decision layer). Finally, the decision is transferred to the response effector. The motor cost asymmetry during the manual response can affect the perceptual decision making process in several different ways. (A) The motor cost for the manual response may only bias the decision layer that involves this response, but leave the decision layer for different response effectors unaffected. If this is the case, the bias observed during the manual response should not generalise to the verbal response. (B) The motor cost may bias the decision layer in general or (C) the sensory representation directly. In either of the latter two cases, the effect of motor cost should be also observable during the response using the different effector.

induced by the motor cost is affecting the decision regardless of the response effector, the vocal decision should be also biased towards the same direction as the manual decision.

For the manual task, the result of Experiment 1 was replicated. The exposure to the resistance made the PSE to shift significantly away from the stimulus associated with the costlier movement (baseline vs. test, mean: −7.06%, paired t-test:$t_{13}$ = 2.94, p=0.012, d = 0.78, 12/14 individuals showed the effect) (*Figure 1E*, *Figure 1—source data 1*). More importantly, for the vocal task, judgement also shifted to the same direction as the manual task (baseline vs. test, mean: −3.00%, paired t-test:$t_{13}$ = 2.44, p=0.030, d = 0.65, 12/14 individuals showed the effect) (*Figure 1E*, *Figure 1—source data 1*), even though the motor cost for the vocal responses was not manipulated. Since the direction of manual motor cost was counterbalanced across participants, this finding cannot be explained by any time-dependent drift of the decision towards one of the directions. This result suggests that the bias induced by the manual motor cost transfers to decisions expressed with other effectors.

Although in Experiment 3, the response effector differed between the manual and the vocal task, the abstract response code ('left'/ 'right') remained the same between the two tasks. Therefore, it is possible that the manual motor cost got associated with these semantic labels, but did not necessarily influence the stimulus-based perceptual decision itself. To test this possibility, in Experiment 4, we again examined the manual-to-vocal transfer of effect caused by the motor cost, but this time varied not only the response effector, but also the response codes between the two tasks. Twelve new participants performed visual motion judgements, where in the baseline and the test phase, manual decisions and the vocal decisions alternated in a mini-block of 11 and 7 trials, respectively (*Figure 3A*). The induction phase involved only the manual task, with gradually increasing left hand resistance. As in Experiment 3, the manual task was a left-right motion *discrimination* task. The vocal task, however, was changed to the motion *detection* task. Participants were asked to detect a near threshold coherent motion by vocally responding 'yes' or 'no'. The to-be-detected motion direction (left or right) was instructed at random before each trial (*Figure 3—figure supplement 1B*). Half of the trials included left or right coherent motion, and in the other half, the coherent motion was absent (0% coherence).

For the manual task, a significant shift of PSE was observed again, reflecting the avoidance of the costly decision (baseline vs. test, mean: −4.39%, paired t-test: $t_{11}$ = 2.88, p=0.015, d = 0.75, 9/12 individuals showed the effect) (*Figure 1E*, *Figure 1—source data 1*). For the vocal task, participants' judgement criterion for leftward motion detection became more conservative after being exposed to the manual motor cost, which was not the case for the rightward motion. The interaction between the phase of the experiment (baseline/test) and the visual motion direction (left/right) was significant, ($F_{1,11}$ = 6.76, p=0.025, $\eta^2$=0.36) (*Figure 3B*, *Figure 3—source data 1*). Since the manual task required a left-right decision and the vocal task a yes-no decision, the abstract response code of these two task were different. Therefore, significant manual-to-vocal transfer cannot be simply explained by the motor cost inducing a bias for choosing a particular type of abstract response label. Instead, the results indicate that the motor cost influenced the perceptual decision – that is the decision based on the feature of the visual stimulus input – itself.

Together, these results demonstrate that the motor cost on the downstream response can recursively change how the input visual stimulus is transformed into the decision; at the level of sensory representation or at the decision layer. In contrast to the criterion, the sensitivity (d') for the motion detection did not change for either visual motion direction ($F_{1,11}$ = 0.44, p=0.52,$\eta^2$=0.04) (*Figure 3—figure supplement 1C*). This indicates that the motor cost did not increase or decrease the signal to noise ratio (gain) of the motion signal to one specific direction.

Until now, we have shown that the motor cost can bias the decision based on a visual stimulus independent from the response effector or abstract response code. Finally, using a model-based approach, we tried to elucidate the processing stage in which the motor cost could have influenced the decision. We analysed both reaction time and choice data of the manual tasks (Experiments 1, 2, 3 and 4; n = 45) under the framework of diffusion decision model (DDM) (*Ratcliff and McKoon, 2008*). The DDM postulates that a decision variable temporally accumulates sensory evidence in favour of one decision (by increasing its value) or in favour of the alternative decision (by decreasing its value). When the decision variable hits a certain threshold level (decision bound), the decision is made and the response is triggered (*Ratcliff and McKoon, 2008*; *Palmer et al., 2005*) (*Figure 4—figure supplement 1A*). Under this framework, we examined whether the source of the manual

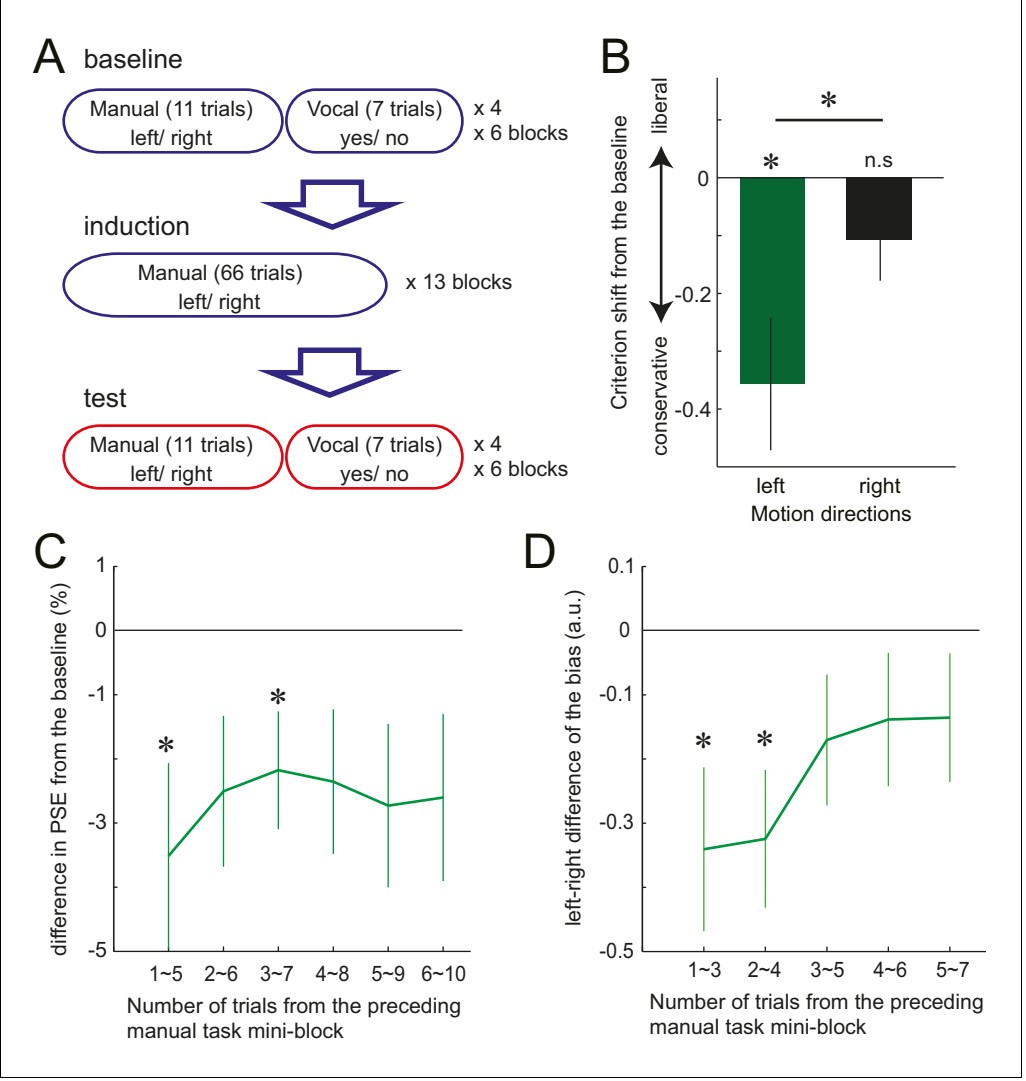

**Figure 3.** Trial structure of Experiment 4 and the effect of preceding motor cost experience on participants' motion judgements. (A) In Experiment 4, participants made vocal judgements to a yes-no motion *detection* task, and manual judgement to a left-right *discrimination* task. (B): Shift of the criterion of motion detection from the baseline during the vocal judgement task in Experiment 4 (d' data is presented in the *Figure 3—Figure Supplement 1C*). Negative value indicates the shift towards more conservative criterion for the motion detection. (C) PSE shift from the baseline condition in Experiment 3, plotted against the number of trials from the preceding manual judgements. Negative value indicates the shift of PSE towards the motion direction with resistance (i.e. decreased judgements towards the motion direction having resistance in their manual response). (D) Vocal motion detection criterion differences between the leftward (with manual response resistance) and rightward (without resistance) motion (Experiment 4). The difference is plotted against the number of trials from the preceding manual judgements. Negative value indicates a more conservative criterion for leftward than for rightward motion. Error bars indicate standard error of mean across participants. Data for *Figure 3B–D* is available as *Figure 3— source data 1*. *p<0.05.

The following source data and figure supplement are available for figure 3:

**Source data 1.** Individual criterion shift of the vocal trials (Experiment 4; summarised in *Figure 3B*), individual PSE shift for the vocal trials across different trials (Experiment 3; summarised in *Figure 3C*) and individual criterion shift for the vocal trials across different trials (Experiment 4; summarised in *Figure 3D*).

**Figure supplement 1.** Task instruction of Experiment 3 and 4, and the d' data of Experiment 4.

decision bias that transferred to the vocal decisions occurred in the sensory representation of the stimulus that is accumulated (*Figure 2C*; sensory representation) or in the decision bound that is used to make the decision (*Figure 2B*; decision layer) (*Ratcliff and McKoon, 2008*; *Palmer et al., 2005*; *Hanks et al., 2006*).

If the bias is introduced at the sensory representation stage, it would increase the input signal (the perceived motion coherence) in the easier direction. We exclude the possibility that the motor cost made the sensory representation of the preferred direction more accurate (increased gain of the signal only in one direction) as we did not observe the discrimination sensitivity change (JND: just noticeable difference) between the baseline and the test condition across different experiments ($t_{44}$ = 0.26, p=0.77). Rather assumed that the motor cost would shift evidence accumulation towards the easier direction (sensory evidence model; *Figure 4—figure supplement 1B*). With this bias, the decision variable would drift towards the preferred decision even in the absence of any coherent visual motion.

Alternatively, we considered the possibility that the motor cost changed the decision bounds (decision layer), that is, the amount of evidence required for each of the choices. This change can be parsimoniously modelled as shift of the starting point of the accumulation process, which will consequently change the distance from the starting point to each decision bound (starting point model; *Figure 4—figure supplement 1C*). The sensory evidence and starting point models predict qualitatively similar pattern of choice probabilities (i.e. bias towards the direction to avoid the motor cost), but different pattern of decision times for correct trials across different motion intensities (Methods, *Figure 4—figure supplement 1B–C*). Therefore, by comparing whether which of the two models explains our data better (*Hanks et al., 2006*; *Ding and Gold, 2012*) (see Materials and method), we may infer the source of the bias.

Additionally to the starting point model and the sensory evidence model, we also fitted a model that allowed for both shifts simultaneously (full model). This allows us to directly compare the effect of each parameter. Also, to check whether the starting point or the sensory evidence shift was necessary to explain the data in the first place, we also prepared a baseline model which we did not model the starting point and/or the evidence accumulation shift, but only modelled the difference in non-decision time (baseline model: see Material and methods). Note that, since we did not record the reaction time of the vocal decisions, this analysis was restricted to model the bias during the manual decisions.

First, we fit each model to the average group data and compared the BIC weights by converting the Bayesian Information Criterion (BIC) for each model (*Wagenmakers and Farrell, 2004*). We then repeated this process 10,000 times, each time drawing 45 participants from our sample with replacement to obtain an estimate of the reliability of our conclusion. The results (*Figure 4C–D*, *Figure 4—source data 1*, *Table 1*) clearly indicate that the starting point model explained the data substantially better than the other models.

Second, we compared the model parameter of the full model fitted to each participants' individual data. Consistent with superior fit of the starting point model, we found a significant shift of the starting point (median; 5.6%, signed rank test: $z_{44}$ = 2.50, p=0.01, d = 0.32; *Figure 4A*, *Figure 4—source data 1*), but no significant change in the evidence accumulation (median; 1.38%, signed rank test: $z_{44}$ = 1.15, p=0.25, d = 0.21; *Figure 4B*, *Figure 4—source data 1*). Therefore, our data suggest that the motor cost biased the decisions by changing the decision layer (starting point) that transforms the input signal into the decision.

The DDM also contains a parameter that captures the motor response time that is independent from the decision (non-decision time; see Material and methods). In the baseline phase, there was no significant difference between the non-decision time of the hands (462.8 ms vs. 468.8 ms, paired t-test: $t_{44}$ = 0.94, p=0.34, d = 0.08). However, in the test phase, although the non-decision time became shorter for both the hands without resistance (dTA: −7.69 ms) and the hand with resistance (dTB: −27.7 ms), the decrease was larger for the hand with resistance (paired t-test: $t_{44}$ = 2.69, p=0.01, d = 0.64). This finding resembles a previous report (*Ding and Gold, 2012*), which demonstrated that the stimulation of the caudate neuron with visual motion directional tuning biased monkey's motion direction judgements towards the neuron's tuned direction, but at the same time decreased the estimated non-decision time for the response (eye movement) to the opposite (non-stimulated) direction.

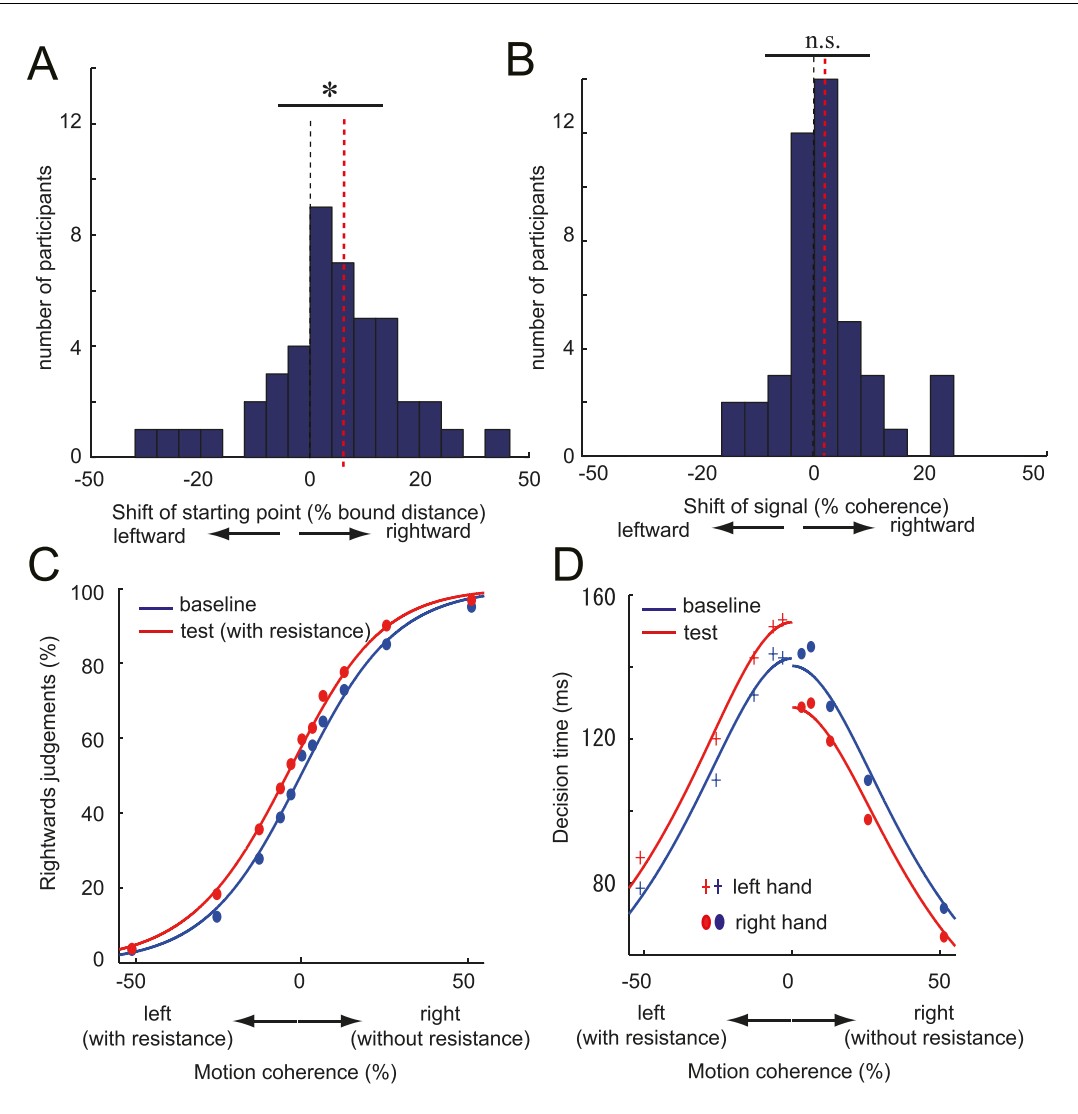

**Figure 4.** DDM parameter estimates and the fitted psychometric and chronometric function. (**A**,**B**) Histogram of individual starting point shift (**A**) and the evidence accumulation shift (**B**) calculated from the DDM (full model). Black dotted line indicates the 0% point (i.e. no effect), and the red dotted line indicates the median of the distribution (i.e. amount of shift). Significant rightward shift of the starting point was observed (median: 5.6%), whereas not for the evidence accumulation shift (median: 1.39%). (**C, D**) Fit of DDM to the choice (**C**) and the decision time (**D**) data averaged across participants (see Materials and methods and *Figure 4—figure supplement 1* Panel **C**). Data for *Figure 4A–D* is available as *Figure 4—source data 1*. *p<0.05.

The following source data and figure supplements are available for figure 4:

**Source data 1.** Individual starting point shift in the test phase from the baseline phase during the manual trials, estimated from the full-model DDM (summarised in *Figure 4A*), individual evidence accumulation shift in the test phase from the baseline phase during the manual trials, estimated from the full-model DDM (summarised in *Figure 4B*), data points consisting the psychometric function estimated from the DDM (starting point model) using the group averaged data (summarised in *Figure 4C*) and data points consisting the chronometric function estimated from the DDM (starting point model) using the group averaged data (summarised in *Figure 4D*).

**Figure supplement 1.** Schematic diagram explaining the drift diffusion model (DDM) and the simulated choice and decision time data.

**Figure supplement 2.** Change of the correct RT, error RT and the correct rate from the baseline to the test phase.

While the DDM models were only fit to the choice probabilities (psychometric function) and the RT function of the *correct* trials (chronometric function, see Material and methods), we also checked whether the models could predict the patterns of RTs on error trials. For this purpose, we simulated individual trials using the estimated group parameters based on the starting point and the sensory evidence models (*Table 2*). For the correct trials, both model simulations showed similar tendencies; the RT reduced for the non-costly motion stimulus compared to the costly stimulus (*Figure 4—figure supplement 2A–B* left panel). The pattern of the error trial RT differed between the two simulations. For the starting point model, error RTs were *shorter* for the costly motions (non-costly decision), whereas the pattern was opposite for the sensory evidence model (*Figure 4—figure supplement 2A,B*). This is because, for the former, the distance between the starting point and the non-costly decision bound decreases, whereas for the latter, the drift rate increases towards the error decision direction for the costly stimulus (i.e. non-costly decision) (*Mulder et al., 2012*). The pattern of RTs for the experimental data (*Figure 4—figure supplement 2C*) was qualitatively similar to that of the starting point model. Therefore, the general pattern of error RTs supported our claim that the motor cost induced a starting point shift.

We showed that participants incorporate the cost of the response into the perceptual decision, and flexibly changes the way of interpreting the sensory environment. To further investigate the temporal dynamics of these flexible changes, we examined how the induced bias developed over the course of the 10 (Experiment 3) or 7 (Experiment 4) trials of vocal decisions following a series of manual decision trials (see detail in the Materials and methods). In Experiment 3, on average, PSE shift was slightly stronger for the vocal trials that immediately followed the manual task (mean shift of PSE; −3.51% for the first five trials (initial point of *Figure 3C*), >−3% for the rest, *Figure 3— source data 1*), although this time dependence did not reach significance ($F_{5, 65}$=0.60, p=0.75). In Experiment 4, the strength of bias (criterion shift) significantly decayed depending on the number of trials from the manual task, showing a stronger bias in the first 4 out of 7 trials (initial two time points of the curve. *Figure 3D*, *Figure 3—source data 1*, $F_{4,44}$ = 2.70 p=0.042, $\eta^2$=0.2). These results indicate that the biasing effect may be relatively short-lived. However, the time scale of the retention is comparable to common perceptual adaptations, such as motion aftereffects. Mather et al. (*Mather et al., 2008*) shows that on maximum, the motion aftereffect lasts for 10 ~ 15 s. A single trial of our task takes at least, 3 ~ 4 s, so our effect lasted for 9 ~ 16 s. Therefore, our results indicate that, in the absence of any further confirmatory evidence of asymmetric response costs for the decision, the brain readapts relatively quickly to the new situation, resembling other examples of spontaneous decay in perceptual and motor adaptation phenomena (*Mather et al., 2008*; *Webster, 2011*; *Smith et al., 2006*). This shows that while perceptual decisions can be updated relatively quickly and flexibly, they do exhibit a substantial memory of past motor costs.

## Discussion

In this study, we showed that visual motion direction decisions can be biased by the cost of the action that is used to report the decision (the *motor cost*). Moreover, we demonstrated that the motor cost indeed affects the decision about the input stimulus identity, and not only the decision about which action to select.

Previous behavioural studies have shown that the perceptual decisions can be biased by changing the frequency (expectation) of stimulus presentation, or by manipulating the response-reward association for the correct/incorrect decisions (*Mulder et al., 2012*; *Whiteley and Sahani, 2008*). Here, we demonstrate that motor costs associated with the response can also bias the perceptual decisions, even when the response is made with a completely different effector that is not associated with increased motor cost (here, verbal instead of manual responses). Therefore, our study provides evidence that the cost on the response for perceptual decisions, which has been regarded as downstream output channel of the decision, can *recursively* influence the decision of the input stimulus itself. In other words, the observed bias not only reflects the *serial* integration of the transient motor cost into the ongoing decision (*Burk et al., 2014*), but represents a more global change in the way of transforming the sensory input to the decision, by taking the prior experience of the motor cost into account (*Makin et al., 2010*).

Congruent with our results, a recent study has shown that asymmetric biomechanical costs induces a bias into the decisions, and that this bias cannot be explained by strategically choosing the

**Table 1.** BIC and BIC weights calculated for different DDM. *BIC and the BIC weights for different DDM models. Values calculated using the group averaged data, and the 95% confidence interval is calculated from the 10,000 bootstrap resampling is presented.

|  |  | Starting point model | Sensory evidence model | Full model | Baseline model |
|---|---|---|---|---|---|
| BIC (95% confidence interval) | averaged | 346.09 | 360.37 | 348.76 | 432.88 |
|  | upper bound | 433.94 | 447.89 | 441.07 | 547.49 |
|  | lower bound | 335.04 | 348.64 | 337.62 | 401.31 |
| BIC weight (95% confidence interval) | averaged | 0.7911 | 0.0006 | 0.2083 | 0.0000 |
|  | upper bound | 0.8600 | 0.4303 | 0.7445 | 0.0000 |
|  | lower bound | 0.0600 | 0.0000 | 0.1336 | 0.0000 |

easier option when perceptual uncertainty is high (*Marcos et al., 2015*). The critical contribution of our study is to show that this influence on the decision process is not limited to the judgements involving the asymmetric motor cost, but generalised to judgement using vocal responses without motor cost manipulation. The present study parsimoniously shows that the motor cost influence is not simply due to the bias of decision at the motor preparation/execution stage.

Our DDM analysis readily explains the effect of motor cost as the change in the required amount of input evidence (i.e. shifts of the starting point) for the decision. Even though only fitted to the reaction times for correct trials (*Palmer et al., 2005*), this model also correctly predicted the RT pattern for error trials. These findings are consistent with previous literature that shows that increasing the presentation frequency or amount of reward for one of the choice biases decision in a way that can be best modelled as a starting point shift (*Mulder et al., 2012*). Indeed, it has been suggested that shifting the starting point of accumulation process is the optimal solution to account for such contextual changes (*Bogacz et al., 2006*; *Simen et al., 2009*). It should be noted, however, that alternative models involving collapsing bounds may perform better in situations in which the stimulus strength varies randomly (*Hanks et al., 2011*; *Tajima et al., 2016*, but also see *Hawkins et al., 2015*).

Electrophysiological studies have shown that the electrical stimulation of parietal or basal ganglia neurons can bias perceptual decisions (*Hanks et al., 2006*; *Ding and Gold, 2012*). These effects were explained by shifts of the starting point in the DDM framework (or equivalently the decision bounds), thus the change in the decision layer of the perceptual task. Therefore, these brain regions are likely candidate neuronal substrates where the motor cost interacts with the sensory input to bias the perceptual decision. Neurons in the lateral intraparietal area (LIP) code the feature that is relevant for the visual decision, independent of response type (*Bennur and Gold, 2011*). Experiments 3 and 4 similarly suggested motor-induced changes for the perceptual decision is independent of the effector used for response. The subcortical network in the basal ganglia has been suggested to represent the cost of action or the 'vigour' of movement initiation (*Mazzoni et al.,*

**Table 2.** Parameter estimates for data from Experiment 1-4 for the starting point and sensory evidence model. *Parameters for the simulations for the error RTs were chosen to be these fitted parameters.

| DDM parameters | $k$ | $A$ | $B$ | $T01(dTA)$ | $T02(dTB)$ | $sp$ | $dcoh$ |
|---|---|---|---|---|---|---|---|
| starting point model | 0.29 | 11.76 | 12.02 | 460 (−9) | 459 (−28) | −1.62 | 0 |
| sensory evidence model | 0.29 | 11.59 | 12.20 | 460 (−17) | 458 (−22) | 0 | 4.00 |

*2007*). Prolonged exposure to altered motor costs during perceptual decisions may similarly change the response properties of these areas, altering how the system judges the sensory evidence from the environment.

How does our current finding relate to the existing theories of perceptual decision making? One of the recent theories is the intentional framework (*Shadlen et al., 2008*). This framework posits that decisions and actions are tightly coupled, with each decision maker separately accumulating the sensory evidence until the threshold level for the specific action is reached. In this scenario, any decision bias induced by imposing a motor cost to a specific action would not transfer to a decision performed by a different action, as there is no explicit communication between the multiple decision makers. Thus, our results indicate that perceptual decisions are either made centrally by a high-order process that is common across different actions (*Filimon et al., 2013*; *Sun and Landy, 2016*), or at least that different local decision makers exhibit a certain degree of mutual dependency, such as a shared cost (value) of the input stimuli (external environment).

In conclusion, we demonstrate that the motor cost involved in responding to a visual classification task is integrated into the perceptual decision process. Our everyday perceptual decisions *seem* to be solely based on the incoming sensory input. They may be, however, influenced by the preceding history of physical cost of responding to such input. The cost of our own actions, learned through the life-long experience of interacting with the environment, may partly define how we make perceptual decisions of our surroundings.

## Material and methods

### Participants

A total of 52 participants (Experiment 1: 12 (6 females), Experiment 2: 10 (5 females), Experiment 3: 16 (8 females), Experiment 4: 14 (5 females); with ages ranging from 18 to 38 years (M = 25.5) participated in the study. All had normal or corrected-to-normal vision, were right-handed and naive regarding the experimental purpose. None of them declared any history of neurological diseases. All participants gave informed written consent, and all procedures were approved by the UCL ethics committee. No statistical test was run to determine sample size a priori. The chosen sample sizes are similar to those in previous publications related to perceptual decisions (*Hagura et al., 2012*; *Rahnev et al., 2011*). Furthermore, we replicated the result of Experiment 1 in the subsequent Experiments 2, 3 and 4 using the similar sample sizes.

### General apparatus

Participants were seated comfortably in front of a virtual environment setup, which has been described in more detail previously (*Reichenbach et al., 2013*). The visual stimulus was presented on the display, which was mounted 7 cm above the mirror. The mirror was mounted horizontally above the manipulanda, preventing direct vision of the hands but allowed participants to view a visual scene on the monitor. During the task, participants leaned slightly forward with their forehead supported by a forehead rest, maintaining the distance from the eye and the mirror constant (25 cm). As a result, the viewing distance from the eye to the monitor was 32 cm. The chair was placed at the position where the participants could most comfortably perform the reaching movement using the manipulanda. Depending on the judgement of the visual stimulus (see below), they made 15 cm straight reaching movements while holding onto a robotic manipulandum (update rate 1 kHz, recording of position and force data at 200 Hz) using their left or right hand (*Figure 1A*). The hand positions were represented by white circles (cursors, 0.3 cm diameter) located vertically above the real positions of the hands. The movement were executed from a starting box (unfilled white squares, 0.5 cm size, 6 cm to the left and right from body midline) to a target box (unfilled white squares, 1 cm size).

### Visual motion stimulus

In the centre of the screen, random-dot motion stimulus was presented (*Britten et al., 1992*) (*Figure 1A*). In a 9 deg diameter circular aperture, dots were presented in a density of 1.7 dot/deg$^2$. The speed of the dots was 10 deg/s. For each trial, 0%, 3.2%, 6.4%, 12.8%, 25.6%, or 51.2% of the

dots moved coherently to the left or the right. All other dots moved in a random direction, picked for each dot separately between 0 and 360 deg.

## Experiment 1

### Task and the movement practice

The trial started with the participants moving the two cursors into the starting boxes. After a delay of 800 ms, a random-dot stimulus was presented. Participants were instructed to judge the direction of the visual motion (left or right), and to make a ballistic reaching movement to the target with either hand. The left judgement required left hand movement, and the right judgement required right hand movement. Initiation of the hand movement made the dot-motion stimulus disappear (*Resulaj et al., 2009*). The stimulus also disappeared if no response had been made after 750 ms from the stimulus presentation. Participants were asked to start moving (make their decisions) as quickly as possible, but before the stimulus disappeared. After the movement, the hands were automatically pushed back near to the starting boxes.

To maintain stable movement kinematics throughout the experiment participants underwent three different types of practice sessions before the main experiment. First, participants responded to a series of 100% coherent leftward or rightward dot motion trials. Participants were asked to perform their reaching movement with a peak velocity of >40 cm/s, and land the cursor within 1.5 cm from the target. When a trial fulfilled this criterion, the visual target 'exploded', informing the participants about the success of the movement. Each training block consist of 48 trials, and the participants continued this training until their success rate exceeded 65% within a block. Next, participants performed three blocks of the same task but using the graded coherence levels as used in the main experiment (66 trials each). Participants were particularly instructed to initiate the response before the stimulus disappeared. Feedback information about the movement kinematics was also presented (see above). Participants were clearly informed that the feedback was *not* about whether the motion direction judgement was correct or incorrect, but about whether their motor performance matched the requirements. Finally, participants performed three blocks of the same task without movement feedback. Participants who could not perform the movements according to the speed criterion did not proceed to the main experiment (two participants).

### Structure of the experiment and the resistance control

There were three phases in the main task; the baseline phase, induction phase and the test phase. Participants performed the same motion direction judgement throughout the experiment, but the resistive force they were exposed during each of the hand movement was different between the phases.

Each phase consisted of 5, 15, 5 blocks of trials, respectively. Each block contained 66 trials, and each the 11 movement direction x coherence level combinations was repeated six times in each block.

The resistive force (f) was velocity dependent, calculated as from the equation;

$$[f_x; f_y] = -\alpha [v_x; v_y]$$

where $v$ denotes for the movement velocity, and $\alpha$ denotes for the coefficient of the viscosity (Ncm/s). Here, negative value indicates the force against the movement direction.

In the baseline phase, the coefficient was set to 0.10 Ns/cm for movements of either hand. In the induction phase, the resistance increased by 0.0008 Ns/cm for each left hand movement. The strength increased until the coefficient reached 0.18 (Ns/cm), and this value remained for the rest of the induction phase and the test phase (*Figure 1B*; upper panel). Progression of the actual peak value of the resistive force (N) through the experiment is presented in *Figure 1—figure supplement 1*.

Our aim of gradually increasing the resistance force was to make the resistance implicit as possible to the participants, avoiding any cognitive strategy to be involved when performing the task. After the experiment, we assessed their awareness using three questions. We first asked participants '*whether they had realised any change in the task during the experiment*', and then more explicitly '*whether they had realised that the resistance increased for either of the hand*'. Only the participants clearly stating '*no*' to both of the questions were included in the analysis. Two participants who

clearly realised the increased left hand resistance (answer 'yes' to both of the questions) were excluded from the analysis. The same procedure was adopted for the remaining experiments; 1, 2, and 2 participants were excluded from the analysis of Experiments 2–4, respectively.

## Analysis

Movement onset was defined as the point when the movement velocity exceeded 2.5 cm/s. Reaction time was defined as the time elapsed between the onset of the visual stimulus to movement onset. Reaction times smaller than 100 ms or larger than 850 ms were excluded from the analysis, since the former decision is unlikely to be based on the visual motion, and the latter is likely to be made after the stimulus disappearance. Movement end was defined as movement velocity falling below 2.5 cm/s. If both hands moved, the hand with the larger movement amplitude was taken as the participant's decision (leftward or rightward) on that trial. Probability of detecting a slight movement for the non-judged hand was 1.9% of all the trials across four experiments. When the movement amplitude of each judged and the non-judged hand was calculated, in Experiment 1, the average amplitude was 15.4 cm and 3.2 cm, respectively. For the movement of the non-judged hand, the movement amplitude that exceeded 2SD from the mean movement amplitude was five trials for one participant, one trial for two participants and 0 trial for the rest of the participants. We confirmed that excluding these few trials did not affect any of the subsequent analysis performed on this data. Same confirmation was also done for Experiment 2 (judged; 13.93 cm, non-judged; 0.23 cm; above 2SD: three trials for one participant, two trials for one participant, one trial for one participant), Experiment 3 (judged; 15.5 cm, non-judged; 2.1 cm; above 2SD: 0 trial for all of the participants) and Experiment 4 (judged; 15.3 cm, non-judged; 2.6 cm; above 2SD: 0 trial for all the participants).

For each participant, the percentage of 'right' judgement responses for each visual motion coherence level was calculated. Logistic regression was used to describe the function of participant's response against the motion strength. The point of subjective equality (PSE), i.e. the motion coherence level at which the participant answered 'rightward' in 50% of the trials was estimated from each regression. This was done independently for the baseline and the test phase, and the PSEs between the two phases were compared using paired t-test (two-tailed). As for all other statistical comparisons, Levene's test was used to confirm the equality of variance before performing this statistical comparison. We additionally confirmed that all the results from the above parametric tests can be replicated by the non-parametric Wilcoxon's signed rank test, which shows that our data is not biased by the particular statistical test used to assess the results.

## Experiment 2

The experiment was largely similar to Experiment 1, except that during the induction phase, reaching movements were not associated with any visual motion judgements. Participants were required to make a left or right hand reaching, according to the arrow presented in the centre of the screen, which pointed either to the left or to the right. Otherwise, the procedure was equivalent to Experiment 1.

## Experiment 3

The structure of the experiment was similar to Experiments 1 and 2; where baseline phase was followed by an induction phase, and finally with the test phase. In the induction phase, the resistance for one of the hands slowly increased while the participants performed the manual perceptual decision about the dot motion (moving left hand for leftward motion, and right hand for right motion). The resistance increased on the left hand for half of the participants (7) and on the right hand for the other half, aimed to account for any hand-dependent effects. For the dot motion, 10 different strengths (i.e. coherence level of the motion) were used (±3.2%, ±6.4%, ±12.8%, ±25.6%, ±51.2%; negative value indicates leftward motion, and the positive for the rightward motion). The induction phase involved 14 blocks of 60 trials each.

During the baseline and the test phase, participants alternated between responding to the visual motion manually (manual task) or vocally (vocal task). During the manual task participants moved their left or right hand according to their perceived motion direction. For the vocal task, participants were asked to indicate the direction vocally without moving their hands. The vocal task started with a tone. After 1500 ms, a random-dot motion stimulus was presented for 500 ms. Participants were

asked to judge whether they perceived a motion direction towards the left or to the right, by vocally answering 'left' or 'right'. Their response was recorded by the experimenter.

Each 10 trials of manual judgements were followed by 10 trials of vocal judgements (mini-block; *Figure 1C*, *Figure 3—figure supplement 1A*). Within a mini-block, the manual and the vocal tasks were presented serially, and this structure was repeated four times within a block (in total, 80 trials per block). Participants performed four blocks each for the baseline phase and the test phase.

### Analysis

The analysis of the manual task was similar to the above experiments. To analyse left and right resistance increase data together, we aligned the data depending on the side of the resistance applied, by assigning negative motion coherence level to the motion direction associated with the direction of the resistance. This is equivalent to converting the right resistance increase data to the left resistance increase data; which was the case for Experiments 1 and 2. The vocal task was analysed similarly to the manual task, in which the PSE between the baseline and the test phase was compared.

To examine the time-dependent effect of the manual motor cost onto the vocal decision, we analysed the vocal task data depending on the number of trials from the last manual trials. To obtain enough trials for the analysis, we calculated the PSE using the first five trials of a mini-block of 10 vocal trials. This procedure was repeated by shifting the window, resulting in six analysis ranges (1st–5th, 2nd–6th, 3rd–7th, 4th–8th, 5th–9th, 6th–10th trials). Finally, PSE of the vocal task during the baseline phase was subtracted from these six values. These values will indicate the change in the strength of the influence of manual motor cost on the vocal decisions over time. We performed a one-way ANOVA to examine this temporal change.

### Experiment 4

The procedure was generally similar to Experiment 3, with the main difference that the vocal task was a motion *detection* task, rather than a motion *discrimination* task as used for the manual task. Also, as in Experiments 1 and 2, the resistance increased for the left hand during the induction phase. The vocal task started with a tone, and the to-be-detected motion direction (left or right) was instructed (*Figure 3A*, *Figure 3—figure supplement 1B*). After 1500 ms, a random-dot motion stimulus was presented for 500 ms. The stimulus included either the near threshold level coherent motion towards the instructed direction or a 0% coherent random-dot motion. Participants were asked to judge whether they perceived a coherent motion towards the instructed direction or not, by vocally answering 'yes' or 'no'. Their response was recorded by the experimenter. The strength of the motion coherence was defined individually before the experiment to approximately match 75% correct rate, causing the percentage correct rate to be between 65% and 85% during the baseline phase of the task.

The aim of the vocal task was to examine whether the bias induced by the motor cost would transfer to the judgement using the different response effector. Additionally, the task was designed such that the abstract response code of the manual task (left-right) would be unrelated that of the verbal task (yes-no). Therefore, the performance of the vocal task could not be biased by the manual task through its commonality of the effector or response code.

Eleven trials of manual task were followed by seven trials of vocal task (*Figure 3A*). This combination of manual/vocal task mini-block was repeated for four times in a block. Therefore, the manual and the vocal tasks were performed serially, similar to Experiment 3. Participants performed six blocks, during each of the baseline phase and the test phase. The induction phase was similar to that of Experiment 1, and contained only the manual tasks, 13 blocks of 66 trials each.

### Analysis

The analysis of manual task was identical to the above experiments. For the vocal task, responses for left motion trials and right motion trials were analysed independently. Any trial in which a hand movement was detected during vocal task was excluded from the analysis. For each motion direction, the sensitivity (d') and the bias (criterion; C) were calculated using signal detection theory (*Macmillan and Creelman, 2005*). Difference of these measures between the baseline and the test phase were compared between the leftward and the rightward motion using two-way ANOVA (phase (2) x motion direction (2)).

We found that the criterion (C) for the leftward motion became more conservative after exposed to increased resistance on the left hand (*Figure 3B*). As in Experiment 3, we also examined whether the strength of this effect decayed as a function of the number of trial since the last manual trial. We calculated the d' and C for both leftward and rightward motion using the first three trials of the vocal judgement in each mini-block (3trials x 24 mini-blocks = 72 trials). We repeated this procedure by shifting the window by one trial, resulting in the five analysis ranges 1st~3rd, 2nd~4th, 3rd~5th, 4th~6th, 5th~7th trial. Then, the values calculated similarly for the baseline (d' and C, each for left-ward and rightward motions) were subtracted, to calculate the change from the baseline condition. Finally, to test for a difference in bias for left and right motions, the left values were subtracted from the right values [i.e. negative values indicate less sensitive (or more conservative) judgements for leftward motion]. Intuitively, this shows how the leftward or rightward bias of d' and the C changes over time as the temporal (trial) distance increases from the preceding manual judgement trials. We performed a one-way ANOVA to examine the temporal change of the left-right bias during the vocal task.

## Diffusion decision model (DDM) analysis

Data of the manual judgement task from Experiments 1, 2, 3 and 4 (n = 45) were re-analysed together under a framework of Diffusion Decision Model (DDM) (*Ratcliff and McKoon, 2008*), to examine the possible source of the decision bias; whether it is (1) increasing the sensory evidence favouring one of the decision (change in the sensory representation; *Figure 2C*, *Figure 4—figure supplement 1B*), or (2) shifting the starting point of the evidence accumulation process more near to one of the decision bounds (equivalent to changing distance to each of the decision bounds) (change in the decision layer; *Figure 2B*, *Figure 4—figure supplement 1C*). For this, reaction time and the choice data of both baseline and the test phase was simultaneously modelled with the DDM, and the estimated parameters were evaluated (*Palmer et al., 2005*; *Hanks et al., 2006*; *Ding and Gold, 2012*). Since we did not obtain the reaction time for the vocal trials, only the manual decision trials across different experiments were analysed. We analysed only the data for non-zero motion coherence level (Experiment 3 did not have 0 coherence level condition) and for the RTs for the correct decision trials, which are established to be well explained by the DDM (*Palmer et al., 2005*; *Shadlen et al., 2006*). The sign of the data from the participants having resistance on the right hand was flipped (Experiment 3), allowing the data to be analysed together with the left hand resistance increased participants.

For the baseline phase, the model had five basic parameters; *A*, *B*, *k*, *T01* and *T02*. In this framework, momentary motion evidence is drawn randomly from a Gaussian distribution N($\mu$, 1), where $\mu$ is calculated as a motion strength (coherence level: Coh) scaled by the parameter *k*: $\mu = k \times$Coh. Decision is transformed into action when the accumulated momentary motion evidence reaches either of the decision bound; *A* (right decision) or –*B* (left decision). Here, leftward decision is the one with the higher resistance for the response. Decision time is defined as the elapsed time between the stimulus onset and the time when the evidence reached either of the decision bound (*Figure 4—figure supplement 1A*). Reaction time is the sum of decision time and the non-decision time (*T01* for a left and *T02* for a right judgement), where the non-decision time is a pure action processing time that is assumed not to depend on the amount of the sensory evidence.

The expected value of rightward judgements across different coherence levels can be calculated as (*Palmer et al., 2005*):

$$\frac{e^{2\mu B} - 1}{e^{2\mu B} - e^{-2\mu A}}.$$

The average decision time for the rightward motion decision can be described as:

$$\frac{A+B}{\mu}\coth(\mu(A+B)) - \frac{B}{\mu}\coth(\mu B),$$

and for the leftward motion decision as:

$$\frac{A+B}{\mu}\coth(\mu(A+B)) - \frac{A}{\mu}\coth(\mu A)$$

To explain the change in decision bias observed between the baseline and test phase within the same model, additional parameters that describe the change in the parameters across two phases (baseline and test) were added to the above five base parameters (delta parameters). Three different models with different delta parameter settings were generated. In the first model (sensory evidence model), the motor cost changed the sensory evidence by changing the motion coherence by *dcoh*. Thus, the motion strength in the test phase was $\mu= k\times$(Coh+*dcoh*). Since we know that the effect of motor cost does not change the discrimination sensitivity (just noticeable difference: JND), but changes only the PSE (*Figure 3C*), change in the sensory evidence is modelled as *addition* to the input stimulus (+*dcoh*), rather than as the change in the gain itself (direct change of *k*). In the second model (starting point model), parameter that indicates the shift of the starting point of the accumulation processes (*sp*) was added, which will consequently change the amount of evidence required for each decision. Equivalently, we can think of this parameter as a shift in the two decision bounds to [*A-sp*] and [–*B-sp*], leaving the distance between the two bounds fixed. In the final model (full model), both coherence level change (*dcoh*) and the starting point shift (*sp*) were added as additional parameters.

In all models across the three models, we also modelled the difference in the non-decision time between the baseline and the test phase. There were 840 (Experiment 3) ~990 (Experiment 1) trials of reaching movement between the baseline and the test phase, and the reaction time is decreased in the test phase compared to the baseline phase regardless of the motion coherence level ($F_{(1,35)}$ =11.95, p=0.0015, $\eta^2$=0.255). We assume that this was due to the reduction of the non-decision time induced by the repetition of the reaching movement. To account for this, we added an additional parameter modelling the decrease of the non-decision time across the two hands. Since such reduction of the non-decision time may differ between the left and the right hand, the difference was modelled separately for the right (*dTA*) and the left (*dTB*). Therefore, the non-decision time for the test phase was modelled as *T01- dTA* and *T02- dTB* for right and left, respectively (same model as ref 17).

As a result, the three DDM models consisted of 8 (sensory evidence model; five basic parameters + *dTA* + *dTB* + *dcoh*), 8 (starting point model; five basic parameters + *dTA* + *dTB* + *sv*), and 9 (full model; five basic parameters + *dTA* + *dTB* + *sv* + *dcoh*) parameters, respectively. In addition to these three experimental models, we also prepared a baseline model, in which we fit the baseline and the test phase data only with the delta parameter of non-decision times (seven parameter baseline model; five basic parameters + *dTA* + *dTB*).

The DDM we used in this study is the most basic one proposed by *Palmer et al. (2005)*. This simple version of the DDM predicts the choice probabilities (psychometric function) and the mean RT function (chronometric function) of the correct trials. Therefore, this model is sufficient to distinguish between the models of interest – a change in starting point of evidence accumulation (*Figure 4—figure supplement 1B–C*). A number of extensions to the DDM framework have been proposed to explain the full RT distributions of correct and incorrect trials using trial-by-trial variability of the drift rate (*Ratcliff and McKoon, 2008*) or by incorporating the time-dependent decision bounds (*Drugowitsch et al., 2012*). While these extensions are important, they do not change the primary predictions regarding the mean RT and choice probabilities under the two models. For the sake of parsimony, we therefore use the simpler model here.

The group-averaged reaction time and choice data of the experiments was first fitted by each of the four models (three models + baseline model), by searching the parameters that minimised the negative log likelihood of the fit (maximum likelihood estimate). We used the group-average data, as each individual had a limited number of trials, and the noise level was rather high. This can induce a bias towards more complex models, as it can over fit the noise. Using group-average data strongly attenuates this effect (*Donchin et al., 2003*; *Thoroughman and Shadmehr, 2000*). To obtain estimates of the reliability of the group-average fit, we resampled the data 10,000 times across participants with replacement, and fit the model to each of the averaged resampled data (*Efron, 1979*). To select the best model to explain the data from the above four, the Bayesian Information Criterion (BIC) (*Schwarz, 1978*) was calculated for each model,

$$BIC = -2\log L + \alpha\log(n)$$

where log*L* denotes for log likelihood of the fitted function, $\alpha$ for number of parameters used for the

fit and *n* for number of data points in the sample. The latter term in the BIC equation penalises the number of parameters used for the fit. Therefore, smallest BIC among the three models will indicate the most parsimonious model. To compare the explanative power between each model in an intuitive way, we converted the BIC values to the BIC (Schwartz) weights (*Wagenmakers and Farrell, 2004*), which expresses the explanatory power of BIC values into ratios among the candidate models.

$$w(i) = \frac{\exp\{-1/2 \triangle BIC(i)\}}{\sum_{k=1}^{K}(\exp\{-1/2 \triangle BIC(k)\})}$$

where *K* is the number of models used to explain the data, $\triangle BIC(i)$ is the difference in BIC from the model with the smallest (best) BIC. The descriptive statistics (averaged and the 95% confidence interval of the 10,000 bootstrap) of BIC value and the BIC weight distributions are summarised in *Table 1*.

We also estimated the delta parameters (*dcoh*, *sp*) of the full model for each individual – thereby avoiding possible biases in the parameter estimates when using averaged data (*Estes and Maddox, 2005*). The parameters were statistically tested against zero (no significant change in the test phase compared with the baseline phase) using a Wilcoxon's signed rank test. The impact of *sp* depends of the distance between the two decision bounds. Therefore, we normalised the individual starting point shift (*sp*) by the estimated distance between the two decision bounds (*sp*/[A + B]).

## DDM simulation analysis for error trial RTs
The parameters of the DDM models were estimated using the proportion of correct decisions and the RT data for correct trials. To test whether this model could also capture the pattern of error RTs, we simulated single trial data from the starting point and the sensory evidence models (10,000 times for each stimulus strength per condition), using the parameters estimated from the group data (*Table 2*). In both models, the leftward judgements is costlier in the test phase. For each simulation, the RT difference between the baseline and the test phase for both correct and error trials was calculated, separately for the leftward and rightward motion stimulus. We also calculated the difference in the correct rates. Then we compared these patterns with the actual experimental data analysed in a same way.

## Acknowledgements
The authors are grateful to Drs. L Ding, T Doi, B Bahrami, the members of the ICN Action and Body Lab and Motor Control Lab for the helpful discussions.

## Additional information

### Funding

| Funder | Grant reference number | Author |
|---|---|---|
| European Research Council | Marie Curie International Incoming Fellowships | Nobuhiro Hagura |
| Japan Society for the Promotion of Science | JSPS Postdoctoral fellowships for research abroad,Kakenhi (25119001) | Nobuhiro Hagura |
| Japan Society for the Promotion of Science | Kakenhi 25119001 | Nobuhiro Hagura |
| Japan Society for the Promotion of Science | 26119535 | Nobuhiro Hagura |
| Economic and Social Research Council | Professional Research Fellowships | Patrick Haggard |
| European Research Council | HUMVOL | Patrick Haggard |
| James S. McDonnell Foundation | Scholar award for understanding human cognition | Jörn Diedrichsen |

The funders had no role in study design, data collection and interpretation, or the decision to submit the work for publication.

## Author contributions

NH, Conceptualization, Formal analysis, Funding acquisition, Investigation, Methodology, Writing—original draft, Writing—review and editing; PH, Conceptualization, Supervision, Funding acquisition, Writing—review and editing; JD, Conceptualization, Resources, Supervision, Funding acquisition, Methodology, Writing—review and editing

## Author ORCIDs

Nobuhiro Hagura, http://orcid.org/0000-0001-9852-5056
Jörn Diedrichsen, http://orcid.org/0000-0003-0264-8532

## Ethics

Human subjects: All participants gave informed written consent, and all procedures were approved by the UCL ethics committee.

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
