## [Decision Letter]

Thank you for submitting your article "Perceptual decisions biased by the cost to act" for consideration by *eLife*. Your article has been reviewed by two peer reviewers, and the evaluation has been overseen by Joshua Gold as the Reviewing Editor and Sabine Kastner as the Senior Editor. The following individuals involved in review of your submission have agreed to reveal their identity: Birte Forstmann (Reviewer #2).

The reviewers have discussed the reviews with one another and the Reviewing Editor has drafted this decision to help you prepare a revised submission.

Summary:

In this elegant study, the authors report four experiments showing that increased motor costs in one decision can influence decision-making in other decisions, even when these other decisions are fundamentally different in effector (voice vs hand) and choice options (detection task vs discrimination task). They report that: 1) increased force for one hand led to biased choices away from that alternative; 2) the biases persisted through vocal choices; and 3) the results could be modelled as a change in the starting point of a drift-diffusion process.

The reviewers agreed that this study is interesting and novel. However, they also believed the manuscript could be improved considerably by addressing several concerns, listed below.

Essential revisions:

1) The authors miss to cite several important references regarding the hypothesis whether the additional cost of selecting one of the motor responses is encoded as a 'starting point shift' or a 'drift rate shift' (e.g., Bogacz et al., 2006; Simen et al., 2009). The authors may also want to consider recent work suggesting that the optimal strategy during trials with different choice payoffs as well as difficulties is to implement an urgency signal (Drugowitsch et al., 2012).

2) The authors fit a simple DDM model to their data to learn more about underlying latent variables. While I think this is a great approach, I have some critical technical remarks about the fitting itself:

a) In general it is not advised to fit cognitive models to data that is collapsed over subjects. It can obfuscate effects and bias towards invalid models (see, e.g., Estes et al. 2005; Brown and Heathcote, 2003). Instead the authors may present the parameter estimates from the model that is fit to individual data.

b) The DDM is not fit separately to different tasks with different effectors. It appears unreasonable to assume that, e.g., the non-decision time parameter is equal for vocal vs. motor responses and I would suggest to fit the DDM separately for each experiment.

c) The authors use a t-test on individual BIC-values for model comparison. A principled way of doing model comparison would be to present the BIC-values in a table indicating which model fit the experiment(s) best. The authors could consider extending such a table by including so-called AIC/BIC-weights which offer a more principled way of comparing BIC values (Wagnemakers and Farrell, 2004).

d) The authors have chosen to fit the most basic version of the DDM without any across-trial variability in non-decision time, drift rate, or starting point. Such a model predicts that error trials and correct trials have exactly the same RT distribution. This is rarely observed in empirical data (Ratcliff and Mckoon, 2008). Why did the authors choose for this simplified model?

3) It is not clear that experiment 2 shows that "mere exposure to an asymmetric motor costs between the two hands is sufficient to bias subsequent decisions" – especially given the relatively rapid, transient effects presented on the detection task (Figure 3). This point should be at least discussed further.

4) Were there changes in the slope of the psychometric function (e.g., in Experiment 1), in addition to the changes in PSE? The somewhat odd, asymmetric effect on detection performance in Experiment 4 might be predicted to have such effects.

5) More discussion should be given to the interpretation of the vocal-transfer effect, and its implication for mechanisms of decision formation – in particular the idea of decisions being formed in the context of an intentional framework or not, which has received much attention in the literature. It also would be useful to discuss this result in terms of what is known about persistent biases in other contexts.

[Editors' note: further revisions were requested prior to acceptance, as described below.]

Thank you for resubmitting your work entitled "Perceptual decisions are biased by the cost to act" for further consideration at *eLife*. Your revised article has been favourably evaluated by Sabine Kastner (Senior editor) and a Reviewing editor.

The manuscript has been improved but there are some remaining issues that need to be addressed before acceptance, as outlined below:

1) It would be useful to provide more details about Experiment 4, including comments about how those details affect the interpretation of the results. In particular, was the induction phase done like in Experiment 3, with half the participants getting increased resistance to the left, and the other half to the right? I don't think that was the case, but unless I missed something the only mention was in the Figure 3 legend ("Vocal motion detection criterion differences between the leftward (with manual response resistance) and rightward (without resistance) motion (Experiment 4)"), which doesn't indicate whether that statement was true just for the Figure 3 data or more generally for Experiment 4.

2) The claim that "it has been suggested that shifting the starting point of accumulation process is the optimal solution to account for such contextual changes [Bogacz et al., 2006 and Simen et al., 2009]" should probably be tempered to include the possibility that these kinds of criterion shifts alone may not be optimal when the signal strength (motion coherence) is randomised; e.g., Hanks et al., J Neurosci 2011.

3) The argument that the simple version of the DDM used in this study can distinguish the models of interest – change in starting point or evidence accumulation – would be stronger if they could show that the manipulations do not generally affect how well those particular model variants fit the data, relative to other models known to more robustly fit choice and RT distributions for these kinds of tasks. For example, are the patterns of error RTs similar in the biased versus unbiased conditions?

---

## [Author Response]

*Essential revisions:*

*1) The authors miss to cite several important references regarding the hypothesis whether the additional cost of selecting one of the motor responses is encoded as a 'starting point shift' or a 'drift rate shift' (e.g., Bogacz et al., 2006; Simen et al., 2009). The authors may also want to consider recent work suggesting that the optimal strategy during trials with different choice payoffs as well as difficulties is to implement an urgency signal (Drugowitsch et al., 2012).*

Thank you very much for the suggestions. We have now incorporated these references in the parts describing the DDM results.

“These findings are consistent with previous literature that has shown that changing the presentation frequency of one of the stimuli or biasing the amount of reward given for the correct trials for one of the choices, biases the participants’ decision pattern in a way that can be best modelled as a starting point shift [Mulder et al., 2012]. Indeed, it has been suggested that shifting the starting point of accumulation process is the optimal solution to account for such contextual changes [Bogacz et al., 2006 and Simen et al., 2009].”

*2) The authors fit a simple DDM model to their data to learn more about underlying latent variables. While I think this is a great approach, I have some critical technical remarks about the fitting itself:*

*a) In general it is not advised to fit cognitive models to data that is collapsed over subjects. It can obfuscate effects and bias towards invalid models (see, e.g., Estes et al. 2005; Brown and Heathcote, 2003). Instead the authors may present the parameter estimates from the model that is fit to individual data.*

Thank you very much for the suggestions. We have now estimated the parameters by fitting the data of each individual participant. We fitted the full model (including both the starting point and the evidence accumulation shift), to directly test whether the motor cost induces significant shifts of starting point or of evidence accumulation. The parameter estimates from the individual fits indeed showed that the parameter of the starting points shift was significantly different from zero (i.e. consistent starting point shift across participants), whereas it was not for the evidence accumulation shift. We also retained our model comparison using BIC. For this purpose, we fit the models to the group-average data, as each individual had a limited number of trials, and the noise level was rather high. High noise levels can lead to erratic fits and unstable answers. Using group-average data strongly attenuates this effect (Donchin et al., 2003; Thoroughman and Shadmehr, 2000). To obtain estimates of the reliability of the group-average fit, we resampled the data across all of the participants (with replacement; 10,000 times) and evaluated the difference in fit across different DDMs. The model fit is now evaluated by calculating the BIC weights (Table 1). We found that the decision bias by the motor cost was best explained by the model that included the starting point shift. Taken together, our data indicates that the motor cost adds consistent directional bias to the starting point, so as to make the decision output biased towards the non-costly decision. Please see subsection “Diffusion Decision Model (DDM) analysis”.

*b) The DDM is not fit separately to different tasks with different effectors. It appears unreasonable to assume that, e.g., the non-decision time parameter is equal for vocal vs. motor responses and I would suggest to fit the DDM separately for each experiment.*

In the vocal decision tasks, we did not obtain the reaction times. Therefore, we fit the model only to data performed for the manual decisions across different experiments. In this case, a combination of participants across experiments is justified, as the manual decision task was always the same. This is now made clearer in the revised manuscript. Please see subsection “Diffusion Decision Model (DDM) analysis”

*c) The authors use a t-test on individual BIC-values for model comparison. A principled way of doing model comparison would be to present the BIC-values in a table indicating which model fit the experiment(s) best. The authors could consider extending such a table by including so-called AIC/BIC-weights which offer a more principled way of comparing BIC values (Wagnemakers and Farrell, 2004).*

Following the suggestions, we have now added a table containing the BIC values and BIC weights, including their upper and lower bound of the 95% confidence interval from the bootstrap in Table 1.

*d) The authors have chosen to fit the most basic version of the DDM without any across-trial variability in non-decision time, drift rate, or starting point. Such a model predicts that error trials and correct trials have exactly the same RT distribution. This is rarely observed in empirical data (Ratcliff and Mckoon, 2008). Why did the authors choose for this simplified model?*

As the reviewer points out, the most basic model that only includes drift rate, threshold value and the non-decision time does not fully explain the mean RT on error trials and the full shape of the empirical RT distribution (Ratcliff and Mackoon, 2008). We used here a DDM as proposed by Palmer et al. (2005), as we only attempted to fit the decision probabilities (psychometric function) and the average RT (chronometric function) of the correct trials. The advantage of this model lies in its simplicity. While adding variability of the drift rate or time-dependent decision bounds could possibly fit the full RT distributions (including those of the error trials) better, these additions would not dramatically change the predictions of either choice probabilities or mean RTs and would therefore not assist in a more accurate choice between the two competing models (motor costs change the evidence accumulation process or the starting value / decision bounds). We have now made this statement more clearly in the revised manuscript. Please see subsection “Diffusion Decision Model (DDM) analysis”.

*3) It is not clear that experiment 2 shows that "mere exposure to an asymmetric motor costs between the two hands is sufficient to bias subsequent decisions" – especially given the relatively rapid, transient effects presented on the detection task (Figure 3). This point should be at least discussed further.*

We conducted this study to explicitly test the idea suggested by one of our previous reviewers; namely, that the bias is induced by the long-term exposure to a direct association between the higher movement cost and a decision. Experiment 2 shows that the direct association during the induction phase is not critical – learning about the movement cost in absence of a direct pairing with the decision seems sufficient. However, you are correct that – given the transience of the effect – the bias could be a result of the asymmetry of resistance during the test phase in Experiment 2, independent of what happens during the induction phase. We have now revised our summary of Experiment 2 as follows:

“This indicates that the direct association of higher motor cost with a specific decision during the induction phase is not critical for inducing the bias. This may suggest that the (implicit) knowledge about the response costs is sufficient to recursively influence the decision. Alternatively, these results could indicate that the bias is only transiently induced during the test phase itself”.

Because Experiment 2 is not critical for our paper, we would be happy to remove it completely from the manuscript. However, we do believe that it is a useful replication of our basic results, while giving some information about the sufficient conditions to obtain the bias.

*4) Were there changes in the slope of the psychometric function (e.g., in Experiment 1), in addition to the changes in PSE? The somewhat odd, asymmetric effect on detection performance in Experiment 4 might be predicted to have such effects.*

When we examined the JND (sensitivity) for the psychometric functions across all experiments, we did not find any systematic difference in the discrimination sensitivity between the baseline and test phase (t44=0.26, p=0.77). Thus, motor cost did not change the perceptual sensitivity, but added a constant bias to the decision. These results justify a critical assumption in our DDM analysis, in which we allowed a bias in the starting value of accumulation or a bias in the amount of accumulation, but not an overall change in the drift rate itself. We have now added this information in the manuscript.

“If the bias is introduced at the sensory representation stage, it would increase the input signal (the perceived motion coherence) in the easier direction. We exclude the possibility that the motor cost made the sensory representation of the preferred direction more accurate (increased gain of the signal only in one direction) as we did not observe the discrimination sensitivity change (JND: just noticeable difference) between the baseline and the test condition across different experiments (t44=0.26, p=0.77).”

*5) More discussion should be given to the interpretation of the vocal-transfer effect, and its implication for mechanisms of decision formation – in particular the idea of decisions being formed in the context of an intentional framework or not, which has received much attention in the literature. It also would be useful to discuss this result in terms of what is known about persistent biases in other contexts.*

Thank you very much for the suggestion. We have now added a paragraph related to this issue as below, which is now placed in the Discussion.

“How does our current finding relate to the existing theories of perceptual decision making? One of the recent theories is the intentional framework [29]. This framework posits that decisions and actions are tightly coupled, with each decision maker separately accumulating the sensory evidence until the threshold level for a specific action is reached. In this scenario, any decision bias induced by imposing a motor cost to a specific action would not transfer to decisions performed by a different action, as there is no explicit communication between the multiple decision makers. Thus, our results indicate that perceptual decisions are either made centrally by a high-order process that is common across different actions [Filimon et al., 2013], or at least that different local decision makers exhibit a certain degree of mutual dependency, such as a shared cost (value) of the input stimuli.”

We also now point to literature showing other decision biases induced by reward pay-off and stimulus probability (Results section paragraph 12).

[Editors' note: further revisions were requested prior to acceptance, as described below.]

*The manuscript has been improved but there are some remaining issues that need to be addressed before acceptance, as outlined below:*

*1) It would be useful to provide more details about Experiment 4, including comments about how those details affect the interpretation of the results. In particular, was the induction phase done like in Experiment 3, with half the participants getting increased resistance to the left, and the other half to the right? I don't think that was the case, but unless I missed something the only mention was in the Figure 3 legend ("Vocal motion detection criterion differences between the leftward (with manual response resistance) and rightward (without resistance) motion (Experiment 4)"), which doesn't indicate whether that statement was true just for the Figure 3 data or more generally for Experiment 4.*

Experiment 4 was performed similarly to Experiment 1 and 2, which the induction phase involved gradual increase of the movement resistance for the left hand. This information was only provided in the Result section. We apologise for the unclear information about the detail. Now, this is also mentioned in the Material and Method section.

*2) The claim that "it has been suggested that shifting the starting point of accumulation process is the optimal solution to account for such contextual changes [Bogacz et al., 2006 and Simen et al., 2009]" should probably be tempered to include the possibility that these kinds of criterion shifts alone may not be optimal when the signal strength (motion coherence) is randomised; e.g., Hanks et al., J Neurosci 2011.*

Thanks you very much for the suggestion. Now we mention about the collapsing decision bound models. Please see Discussion paragraph four.

*3) The argument that the simple version of the DDM used in this study can distinguish the models of interest – change in starting point or evidence accumulation – would be stronger if they could show that the manipulations do not generally affect how well those particular model variants fit the data, relative to other models known to more robustly fit choice and RT distributions for these kinds of tasks. For example, are the patterns of error RTs similar in the biased versus unbiased conditions?*

Thank you very much for the suggestions. We have now looked at the RT pattern of error trials. Figure 4—figure supplement 2 shows the difference in RT between baseline and test phase. RTs were overall faster for the test-phase than the baseline-phase. The RT for correct trials decreased more for rightward stimuli (non-effortful option) than for leftward stimuli. For the error trials, RTs of rightward responses to leftward stimuli decreased more. We then compared this result with simulation from the model with a starting-point shift (Figure 4—figure supplement 2) and with a change in drift rate (Figure 4—figure supplement 2). As can be seen, the actual RT pattern of correct and error trials was similar to that predicted by the starting point bias model simulation, supporting our initial conclusion.

In detail, simulations were performed based on a simple diffusion decision process. The momentary evidence is randomly sampled from the normal Gaussian distribution, where its mean (μ) is the multiplication of the drift gain (k) and the input motion strength (Coh). When the accumulated evidence reaches either of the decision bound, i.e. right decision (A) or left decision (–B), the decision is made. RT was defined as sum of the time for the evidence to reach either of the decision bound from the stimulus onset and the time to move the effector (non-decision time; T01 for right hand, T02 for left hand). For the two simulations, we used the parameters estimated from our experimental data using the analytical DDM solution on the correct RT trials only (Palmer et al., 2005) (see Table 2 for details).

For the starting point bias simulation, the starting point (sp) shifted 6.99% (of the inter-bound distance) to the right. For the evidence accumulation bias, the shift in motion coherence (dcoh) was 4.0% to the right. For both simulations, we confirmed that our model fit can accurately recover the magnitude of these bias parameters (7.0% and 4.33% respectively).

The bias increased the probability of rightward decision for the rightward (non-costly) stimulus, and decreased the probability of leftward (costly) decision for the leftward stimulus (Figure 4—figure supplement 2, right panels). As done for the real data, we then calculated the RT difference between the baseline and the test phase for both correct and error trials. Note that, overall drop in RT is caused by a general drop in the non- decision time from the baseline to the test phase. For the correct trials, RTs became shorter for the non-costly (right) motion compared to the costly (left) motion. However, for the error trials, pattern of RT change were different between the starting point model and the sensory evidence model. For the starting point model, RTs for the error trials became longer for the non-costly (right) motion compared to the costly (left) motion (Figure 4—figure supplement 2). This is due to the starting point becoming distant from the leftward decision bound (the error direction for the rightward stimulus). In contrast, for the sensory evidence model, error RTs for the non-costly motion became shorter than the costly motion (Figure 4—figure supplement 2). This is because, when the leftward (costly) stimulus is presented, the added bias can switch the drift direction, and increase the drift rate for the rightwards (non-costly, error) direction. Therefore, there is a clear qualitative difference in the error RT pattern between the two different types of biases (see also Figure 2 of Mulder et al., 2012).

The data for the error RT looked qualitatively more similar to simulation from the starting point model. In summary, our simple model fit only to the correct RTs can accurately recover the pattern of the error RTs. Taken together, we believe that the perceptual decision bias induced by the motor cost is primarily driven by the shift in the starting point of the evidence accumulation process.

Now these new analysis and data are added to the revised manuscript. Please see Results section and Materials and methods section, Figure 4—figure supplement 2 and Table 2.